# Mitochondrial Dysfunction in the Pathogenesis and Treatment of Oral Inflammatory Diseases

**DOI:** 10.3390/ijms242015483

**Published:** 2023-10-23

**Authors:** Zhili Dong, Liping Wu, Hong Hong

**Affiliations:** 1Hospital of Stomatology, Sun Yat-sen University, Guangzhou 510055, China; dongzhli@mail2.sysu.edu.cn (Z.D.); wulping@mail.sysu.edu.cn (L.W.); 2Guangdong Provincial Key Laboratory of Stomatology, Sun Yat-sen University, Guangzhou 510055, China

**Keywords:** mitochondrial dysfunction, oxidative stress, redox balance, periodontitis pathogenesis, periodontitis treatment

## Abstract

Oral inflammatory diseases (OIDs) include many common diseases such as periodontitis and pulpitis. The causes of OIDs consist microorganism, trauma, occlusal factors, autoimmune dis-eases and radiation therapy. When treated unproperly, such diseases not only affect oral health but also pose threat to people’s overall health condition. Therefore, identifying OIDs at an early stage and exploring new therapeutic strategies are important tasks for oral-related research. Mitochondria are crucial organelles for many cellular activities and disruptions of mitochondrial function not only affect cellular metabolism but also indirectly influence people’s health and life span. Mitochondrial dysfunction has been implicated in many common polygenic diseases, including cardiovascular and neurodegenerative diseases. Recently, increasing evidence suggests that mitochondrial dysfunction plays a critical role in the development and progression of OIDs and its associated systemic diseases. In this review, we elucidated the critical insights into mitochondrial dysfunction and its involvement in the inflammatory responses in OIDs. We also summarized recent research progresses on the treatment of OIDs targeting mitochondrial dysfunction and discussed the underlying mechanisms.

## 1. Introduction

A mitochondrion is a double membrane-enclosed organelle found in most eukaryotic cells, with a size ranging from 0.5 to 10 μm in diameter and the number of mitochondria per cell varying according to energy demand. Mitochondria are often referred to as the cellular power plants or energy center because they generate most of the adenosine-5′-triphosphate (ATP) via oxidative phosphorylation (OXPHOS) as the major source of chemical energy for physiological processes. Since mitochondria generate most reactive oxygen species (ROS) via OXPHOS and possess effective antioxidant systems, they play a central role in regulating oxidative stress and cellular redox homeostasis [1,2]. In addition to generating cellular energy, mitochondria are involved in many other processes, such as lipid metabolism, cellular differentiation, immune regulation, apoptosis, autophagy, cell growth and protein synthesis [3,4,5]. Given that mitochondria play pleiotropic roles in cellular physiology, mitochondrial dysfunction can directly regulate cell and tissue homeostasis and participate in the pathological process of many systemic diseases such as diabetes mellitus, cancer and autoimmune diseases [6,7,8].

Oral inflammatory diseases (OIDs) such as chronic periodontitis and pulpitis are becoming increasingly common, which not only damage orofacial tissue but also contribute to the increased risks of many systemic diseases [9,10]. Since OIDs have both high prevalence and recurrence, and present treatment methods for OIDs are mostly symptomatic treatment, many patients are constantly suffering from the negative impact of OIDs on life quality [11,12,13]. Therefore, it is imperative to explore the underlying causes of OIDs as well as develop new therapeutic strategies. Recently, mitochondrial dysfunction gained increasing attention in the pathogenesis and progression of OIDs by regulating oxidative stress and activating immune responses [14,15,16]. Here, we summarized the latest research regarding how mitochondrial dysfunction contributes to OIDs including periodontitis, pulpitis, osteoradionecrosis of the jaw and Sjögren’s syndrome. We also discussed whether targeting mitochondria could act as a possible strategy for OIDs treatment in the future.

## 2. Materials and Methods

Disrupted mitochondrial homeostasis may damage redox balance and affect normal cellular metabolism, which finally results in pro-inflammatory responses. Recently, accumulating evidence further uncovered the role of mitochondrial dysfunction in the pathogenesis and progression of oral inflammatory diseases such as periodontitis and pulpitis. Databases MEDLINE (through PubMed) and Web of Science were searched for the latest research on mitochondrial dysfunction in OIDs using the keywords “Mitochondrial dysfunction”, “Oral inflammatory diseases”, “Periodontitis”, “Pulpitis”, “Osteoradionecrosis” and “Sjögren’s syndrome”.

## 3. Role of Mitochondrial Dysfunction in Inflammatory Diseases

In the past, mitochondria have been considered to be the center of energy production for cellular energy needs; however, it has become more recognized that mitochondria also play central roles in ROS production, calcium homeostasis, cellular signaling and immune responses [17,18]. Under a pathological context, cell and tissue stress caused by a series of stimulations, including pathogens, senescence and exposure to environmental toxicants, can directly or indirectly lead to mitochondrial dysfunction [19,20,21]. Typical features of mitochondrial dysfunction are impaired respiratory chain function, structural abnormalities, depletion of cell ATP pool, disrupted cellular signaling and increased mitochondria-derived ROS (mtROS) generation [22,23]. Excessive mtROS will induce oxidative stress and cause oxidative damage to the mitochondrial structure and function, forming a vicious cycle [24]. Such stimulations can also disrupt mitochondrial membrane integrity, leading to the release of mitochondrial ligands or damage-associated molecular patterns (DAMPs) [25,26]. Mitochondrial DAMPs include N-formylated peptides, cardiolipin exposure, mtROS and mitochondrial DNA (mtDNA). These molecules then activate pattern recognition receptors (PRRs) of the immune system and trigger a wide array of inflammatory responses, including neutrophil activation, NLRP3 (NOD-, LRR- and pyrin domain-containing 3) inflammasome activation and pro-inflammatory cytokine and chemokine production [27,28]. In addition, mitochondrial dysfunction can also regulate immune responses via directly affecting the metabolism of immune cells, such as T cells, B cells and macrophages [29,30,31,32]. For example, mitochondrial dysfunction induced by excessive uptake of free fatty acids could drive the activation of the NLRP3 inflammasome in macrophages and induce the release of interleukin (IL)-1β [33]. Therefore, mitochondria are essential organelles for the regulation of innate immunity and inflammatory responses against infectious pathogens or in the context of autoimmune diseases (Table 1). In this section, we briefly summarize how mitochondrial dysfunction affects immune responses in different kinds of inflammatory diseases, according to recent studies.

## 4. Mitochondrial Dysfunction in the Pathogenesis, Progression and Treatment of OIDs

### 4.1. Periodontitis

#### 4.1.1. The Role of Mitochondrial Dysfunction in the Etiopathogenesis of the Chronic Periodontitis

As one of the most common human diseases, periodontitis is a chronic inflammatory disease that affects about 45% of adults, rising to over 60% in people aged over 65, which creates a significant healthcare, social and economic burden when left untreated or not treated appropriately [48,49]. Common features of periodontitis include gingival inflammation, clinical attachment loss and alveolar bone loss [50]. Although the main causative factor is microorganisms which colonize the subgingival dental plaque—inducing an exaggerated inflammatory response—genetic predisposition, smoking, poor oral hygiene and malnutrition are also important factors in the pathogenesis and progression of periodontitis [51,52,53,54]. Despite recent advances in the understanding of the pathological process, common treatments for periodontitis, including basic treatment, periodontal surgery and adjuvant drug administration, still provide insufficient periodontal tissue repair [55,56,57]. Recently, many studies have suggested that mitochondrial dysfunction could also contribute to the initiation of periodontitis and increase the risk of its related systemic diseases [52,58,59,60]. Govindaraj et al. conducted a study focusing on mitochondrial dysfunction in the periodontal tissue of chronic periodontitis (CP) patients and found that compared to healthy subjects, mitochondrial membrane potential and oxygen consumption rate of gingival cells from CP patients were reduced by four- and five-fold, respectively, whereas, ROS production was increased by 18%. Moreover, mitochondrial DNA sequencing revealed 14 mutations existed only in periodontal tissues but not in circulation, suggesting that mitochondrial dysfunction and genetic heterogeneity could contribute to the pathogenesis of CP [61]. These studies highlighted the value of mitochondrial function analysis in the early diagnosis of periodontitis.

Periodontal ligament stem cells (PDLSCs) are a kind of somatic mesenchymal stromal cell (MSC) which show typical mesenchymal stromal cell properties, such as self-renewal, multilineage differentiation and immunoregulation, which are essential for homeostasis in periodontal tissue [62]. Notably, PDLSCs are impaired under periodontitis and are involved in the progression of inflammation by aggravating immune response and stimulating osteoclast differentiation [63,64]. Li et al. adopted a quantitative proteomic technique to investigate the protein expression pattern during human PDLSCs osteogenesis. They found that protein related to OXPHOS may be essential in the osteogenesis process, which highlighted the role of mitochondria in regulating PDLSCs’ differentiation ability [65]. Chen et al. found that mitochondria abnormalities are present in the oxidative stress-induced periodontal ligament fibroblast apoptosis, judging by the increased mtROS amounts, upregulated mitochondrial membrane potential and ATP production [66]. Liu et al. showed that LPS-induced inflammatory responses in human gingival fibroblasts (HGFs) were partially dependent on the interaction between P53 and ROS. Upon activation, P53 and ROS formed a feedback loop and led to disrupted redox imbalance and mitochondrial dysfunction in periodontitis, triggering increased secretion of IL-1*β*, IL-6 and tumor necrosis factor (TNF)-*α* [67]. Liu et al. collected a gingiva tissue sample from CP patients and observed greater mitochondrial structure destruction, reduced mtDNA copy and lower mitochondrial protein PDK2 levels, compared with those samples from healthy individuals [68]. Similarly, HGFs from periodontitis patients exhibited increased levels of mitochondrial p53, enhanced mtROS production and secretion of pro-inflammatory cytokines, as compared to HGFs from healthy donors [69].

In an experimental periodontitis rat model, Franca et al. observed morphometric changes in renal tissues and disruption of the brush border in renal tubules, accompanied by an increase in oxidative stress and lipid peroxidation in kidneys [70]. In another study, melatonin treatment in periodontitis rats was found to be effective in restoring redox balance in gingival tissue and reducing oxidative stress levels in circulation, which could alleviate kidney injury [71]. Sun et al. observed that diabetic rats with periodontitis presented more severe mitochondrial dysfunction than non-diabetic rats with periodontitis, reflected by the decreased ATP production, reduced gene expression of electron transport chain complex I subunits and weaker mitochondrial biogenesis. They demonstrated a close correlation between these mitochondrial events and periodontal tissue damage, proving that impaired mitochondrial function contributed to the pathogenesis of periodontitis in diabetic rats [72]. Another study found that diabetic rats displayed enhanced macrophages infiltration and M1 polarization in periodontal lesions, compared with vehicle-treated rats. Under LPS or IL-4 stimulation, RAW264.7 macrophage cells showed elevated ROS levels and increased expression of M1 macrophage markers, which could be reversed by N-acetylcysteine treatment [73].

Atherosclerosis (AS) is a chronic artery disease characterized by plaque formation and chronic vascular inflammation. Many epidemiological studies have described the correlation between periodontitis and carotid AS [74,75]. Porphyromonas gingivalis (*P. gingivalis*), a well-known pathogen in periodontitis progression, has been shown to accelerate lipid droplet accumulation in macrophages, partially through the induction of ROS production, leading to disturbed lipid homeostasis and foam cell formation during AS [76]. *P. gingivalis* infection can also induce mitochondrial fragmentation, disrupt redox balance and decrease ATP concentration in vascular endothelial cells. Researchers suggested that the phosphorylation and recruitment of Drp1, a key protein involved in mitochondrial fission, might be the key events leading to mitochondrial dysfunction in *P*. *gingivalis*-infected endothelial cells, providing new insights into how periodontal pathogen-induced mitochondrial dysfunction exacerbates atherosclerotic lesions [77].

Taken together, the findings mentioned above indicate that mitochondrial dysfunction participates in the pathogenesis and progression of periodontitis by affecting oxidative stress and regulating inflammatory responses (Figure 1). Therefore, developing novel strategies to evaluate mitochondrial function in periodontitis patients may assist the diagnosis and treatment of such disease as well as its complications.

#### 4.1.2. Mitochondrial Dysfunction-Targeted Therapies

Liu et al. identified TRPA1, an important transient receptor potential (TRP) cation channel, as an important factor in periodontium destruction in periodontitis. Inhibiting TRPA1 markedly reduced oxidative stress and apoptotic levels in LPS-treated PDLSCs, via the inhibition of endoplasmic reticulum (ER), and mitochondria stress, via downregulating PERK/eIF2α/ATF-4/CHOP pathways [78]. Periodontitis is closely related to hypoxic microenvironment. Previous studies have demonstrated that oxygen saturation in periodontal microenvironment reduced by 6% [79]. Cementoblasts possess similar characteristics with osteoblasts and generate cementum in the reconstruction process of periodontal tissue [80,81]. Wang et al. showed that overexpression of peroxisome proliferator-activated receptor gamma coactivator-1 alpha (PGC-1α), a critical regulator of mitochondrial biogenesis, can partially reverse the inhibition of cementoblasts mineralization and mitochondrial biogenesis caused by CoCl_2_-induced hypoxia [82].

Since oxidative stress is recognized as one of the key regulators in periodontitis, the therapeutic efficacy of several antioxidants on periodontitis are examined. Zhao et al. showed that rutin treatment inhibited the release of ROS, increased the secretion of antioxidative factors and promoted PDLSCs proliferation via the PIK3/AKT signaling pathway under an inflammatory environment [83]. Similar effects were also observed on hyperglycemic periodontitis rats [84]. Hydroxytyrosol (HT), a natural phenolic compound possessing antioxidative abilities, could inhibit mitochondrial dysfunction by decreasing optic atrophy 1 (OPA1) cleavage and by elevating AKT and GSK3β phosphorylation, which helped prevent oxidative stress-induced osteoblast apoptosis [85]. HT was also reported to exert a therapeutic effect on the periodontitis mice model via repressing RANKL-induced osteoclast maturation and promoting osteogenic differentiation. Such effect was partly dependent on attenuating mitochondrial dysfunction and inhibiting ERK and JNK pathways [86]. In addition to antioxidants, photodynamic therapy (PDT) has shown a protective effect on periodontitis by targeting mitochondria as well. Jiang et al. found methylene blue-mediated PDT could induce macrophage apoptosis in vitro and in rats with periodontitis via regulating ROS levels and reducing mitochondrial-dependent apoptosis, suggesting that the potential of PDT in treating periodontitis does not only rely on its antimicrobial capacity [87,88].

Recently, nanomaterials showed great potential in the field of periodontitis therapy [89,90]. Ren et al. synthesized a nanocomposite with ROS-scavenging activity by combining CeO_2_ nanoparticles (CeO_2_ NPs) onto the surface of mesoporous silica. Periodontal administration of such nanoparticles efficiently reduced ROS levels and improved the osteogenic differentiative capacity of hPDLSCs with H_2_O_2_-induced oxidative stress injury [91]. The same group synthesized controlled drug release nanoparticles by encasing mitoquinone (MitoQ, an autophagy enhancer) into tailor-made ROS-cleavable amphiphilic polymer nanoparticles. Once exposed to ROS under oxidative stress conditions, the ROS-cleavable structure disintegrated, promoting the release of the encapsulated MitoQ. The released MitoQ efficiently induced mitophagy through the PINK1-Parkin pathway and reduced oxidative stress, which contributed to a redox homeostasis and facilitated periodontal tissue regeneration [92]. Qiu et al. fabricated a ROS-cleavable nanoplatform by encapsulating N-acetylcysteine into tailor-made amphiphilic polymer nanoparticles which could decrease osteoclast activity and inflammation in the periodontitis rat model and improve the restoration of destroyed periodontal tissue [93]. Zhai et al. showed that obstructed mitophagy and Ca^2+^ overload led to dysfunctional mitochondria accumulation in MSCs isolated from periodontitis and osteoarthritis patients. They engineered mitochondria-targeting and intracellular microenvironment-responsive nanoparticles to attract Ca^2+^ around mitochondria in MSCs to regulate calcium flux into mitochondria, which successfully restored the mitochondrial function of diseased MSCs and rescued periodontal tissue damage [94].

In addition to alleviating periodontitis-induced periodontal tissue damage, targeting mitochondrial dysfunction in periodontitis also contributes to ameliorating the complications of periodontitis. Li et al. demonstrated that resveratrol could prevent periodontal tissue destruction, as indicated by the improvement in pocket depth, gingival bleeding and tooth mobility. Meanwhile, resveratrol administration also alleviated periodontitis-induced kidney injury by means of decreasing oxidative stress, regulating mitochondrial membrane potential and increasing mtDNA such as Sirtuin 1 and PGC-1α [1]. Febuxostat, a potent xanthine oxidase inhibitor, has been shown to attenuate the progression of periodontitis in rats by reducing inflammatory cytokine levels and oxidative stress. It also attenuated periodontitis-induced glucose intolerance and blood pressure elevations, suggesting its therapeutic potential in treating patients with both diabetes and periodontitis [95].

### 4.2. Pulpitis

Pulpitis is one of the most common OIDs affecting millions of people worldwide and it is a major cause of tooth loss [96]. According to Global Burden of Diseases, Injuries, and Risk Factors Study 2016 data, dental caries was the most prevalent among all diseases and it is recognized as a direct contributor to pulpitis [97]. Bacterial infection and immune responses are closely related to the initiation and progression of pulpitis, so, blocking bacterial invasion and alleviating pulp inflammation at an early stage are crucial for pulpitis prevention and treatment. However, recent studies have revealed the imbalance in mitochondrial dynamics associated with irreversible pulpitis. Vaseenon et al. analyzed mitochondrial quality control-related protein expression in inflamed dental pulp tissue harvested from pulpitis patients. Compared to the control group, dynamin-related protein 1 (Drp1) was significantly higher in the pulpitis group, while mitofusin 2 (MFN2) and OPA1 were significantly lower, indicating a hyper-activated mitochondria fission status [98]. Buzoglu et al. collected samples from healthy donors and pulpitis donors to investigate the difference in oxidative stress cycles. They revealed that caries-related inflammatory response altered the oxidative stress cycle in pulp tissue and GSH (glutathione) levels were upregulated due to the increase in ROS levels, which improved the defensive capacity of the dental pulp [99]. Vengerfeldt et al. found higher levels of oxidative stress in pulpitis, and periodontitis patients were associated with increased dental pain and bone destruction [100]. The results mentioned above reveal how mitochondrial dysfunction and oxidative stress affected pulpitis pathogenesis and its related symptoms.

Pan et al. showed that lysophosphatidic acid protected human dental pulp cell injury from ischemia-induced injury by maintaining mitochondrial membrane potential and preventing mitochondrial-mediated apoptosis [101]. Guo et al. found that saxagliptin can exert a protective effect on LPS-induced damage in dental pulp cells by targeting mitochondrial dysfunction. In brief, saxagliptin ameliorated LPS-induced overproduction of ROS and reduction in glutathione (GSH). Saxagliptin treatment also prevented LPS-induced mitochondrial dysfunction by restoring mitochondrial membrane potential and ATP production [102]. Interestingly, Zhang et al. found that mitochondria from healthy dental pulp stem cells (DPSCs) could transfer to injured DPSCs, after being co-cultured, and promoted functional recovery. An in vivo study demonstrated that transplantation of exogenous dental pulp stem cells can exert a mitochondrial transfer function to repair injured pulp tissue and promote pulp–dentin complex recovery, which can be modulated by Mfn2 expression [103].

Odontoblasts play an essential role in maintaining a stable pulp microenvironment. During the progression of dental caries, odontoblasts can identify and respond to bacteria invasion and form reparative dentin to protect pulp tissue [104]. Zhang et al. revealed that leakage of mtDNA upon LPS stimulation promotes pulpitis progression through GSDMD-mediated pyroptosis. Secretion of the pro-inflammatory cytokines CXCL10 and IFN-β was also induced following mtDNA release [105]. They also identified mtDNA from damaged mitochondria in the cytosol as an activator of the cGAS-STING pathway and IL-6 secretion [16]. They confirmed that odontoblasts could receive exogenous mitochondria through tunneling nanotubes, thereby reducing mitochondrial dysfunction and ROS-NLRP3 inflammasome-triggered cell pyroptosis. When stress conditions occur, pyroptotic odontoblasts would increase TNF-αto-promoted tunneling nanotubes formation via NF-κB signaling, resulting in elevated mitochondrial transfer efficiency, which can be recognized as a self-defense mechanism [106]. To sum up, stabilizing mtDNA and developing effective measures to remove cytosolic mtDNA in time and transferring healthy mitochondria may be important for controlling inflammatory response and preventing irreversible pulp damage.

### 4.3. Osteoradionecrosis

Osteoradionecrosis (ORN), also known as radiation osteomyelitis, is a chronic clinical complication in patients with head and neck cancer following radiotherapy (RT), with an incidence of 5–10% [107]. Symptoms of ORN include pain, trismus, pathological fracture and orocutaneous fistulae, which severely impair patients’ quality of life. Histomorphometric analysis showed that necrotic bone, inflammatory infiltration and reactive bone formation were present in samples from ORN patients [108]. Danielsson et al. found oxidative stress response affected individual radiosensitivity and induced healthy tissue damage following RT [109]. RT can induce high levels of ROS, leading to microvascular structure disruption, ischemia and inflammatory responses [110]. These studies indicate that mitochondrial dysfunction and oxidative stress are extensively implicated in the pathogenesis and development of ORN.

Wang et al. fabricated CeO_2_ NPs and tested the protective effect on MC3T3-E1 osteoblast-like cells treated with X-ray irradiation. Both intracellular ROS production and extracellular H_2_O_2_ concentration increased after X-ray exposure but reduced after CeO_2_ NPs treatment. They suggested that CeO_2_ NPs treatment is effective in reducing osteoblast injury following X-ray irradiation and may be a novel therapy for ORN [111]. Li et al. demonstrated that pretreatment with α2-macroglobulin (α2M) reduced the apoptosis rate and improved the antioxidant capacity in bone marrow mesenchymal stem cells treated with 8 Gy irradiation. In the ORN animal model, α2M administration suppressed 8-hydroxy-2’-deoxyguanosine expression in mandibular bone and tongue paraffin-embedded sections, which is an indicator of oxidative damage, and increased SOD2 expression in mucosa and tongue paraffin-embedded sections. Microstructural analysis showed that mandibular bone loss was alleviated by α2M administration [112]. The studies mentioned above showed that alleviating oxidative stress and mitochondrial dysfunction might be the key to the management of ORN but still requires further clinical evidence.

### 4.4. Sjögren’s Syndrome

Sjögren’s syndrome (SS) is a chronic autoimmune disease characterized by B cell hyperactivity and lymphocytic infiltration of the exocrine glands, especially the lacrimal and salivary glands. The estimated SS prevalence is 0.5–1.5% worldwide and the male to female ratio is about 1:10 [113]. Typical symptoms of SS include keratoconjunctivitis sicca, xerostomia, fatigue and musculoskeletal pain, which negatively affect both the physical and mental health of SS patients. Genetic susceptibility, mental stress and viral infection are recognized as potential contributors to SS, but the mechanism of SS pathogenesis remains unclear [114,115]. Recent studies showed that autoimmune-based mitochondrial dysfunction might be involved in SS pathogenesis. Li et al. observed ultrastructure changes in cellular organelles in both acini and ducts from salivary glands, including the swelling of mitochondria and disrupted membrane integrity [116]. They also identified four mitochondria-related genes (*CD38*, *CMPK2*, *TBC1D9*, *PYCR1*) as a potential link between mitochondrial dysfunction and immune response activation of SS, postulating the significance of disrupted mitochondrial dynamics and impaired respiratory chain stability on SS development [117]. Katsiougiannis et al. collected salivary gland epithelial cells (SGEC) and found that the mitochondrial proteome of SGEC was significantly altered compared to the control group, especially enzymes related to pyruvate metabolism, fatty acid β-oxidation and TCA cycle. Morphological alterations were also captured under transmission electron microscopy, where mitochondria in SS-SGEC displayed strong reduction in cristae abundance, disrupted cristae contour and hypodense matrix [118]. Zhao et al. performed a mtDNA-based analysis among female SS patients to evaluate its association with the development of SS and suggested that single nucleotide polymorphisms (SNPs) in the mitochondrial displacement loop could modify SS progression by regulating inflammatory cytokine expression [119]. Benedittis et al. observed a significant increase in mitochondrial dynamics-related gene expression in SS patients, specifically in mitochondrial fission and fusion homeostasis, such as mitofusin-1 (MFN1), and mitochondrial transcription factor A (TFAM). Their research suggested a possible involvement of mitochondrial dysfunctions in the pathogenesis of SS [120]. Yoon et al. found that mitochondrial double-stranded RNAs (mt-dsRNAs) were elevated in the saliva and tears of SS patients and in the salivary glands of diabetic mice with salivary dysfunction. They constructed a three-dimensional culture of human salivary gland cells and found that mt-dsRNAs were induced by exogenous dsRNAs stimulation. Treatment of dsRNAs also activated the innate immune system, triggered IFN-mediated immune alterations and promoted glandular phenotypes. Direct suppression of mt-dsRNAs reversed the glandular phenotypes of SS [121]. The same group also demonstrated the therapeutic efficacy of muscarinic receptor ligand acetylcholine and resveratrol in reversing the activated response to immunogenic stressors in salivary gland acinar cells [122]. Xu et al. confirmed that lactate levels were significantly upregulated in the salivary glands of patients with SS and lactate triggered an inflammatory response by damaging mtDNA and causing mtDNA leakage in glandular epithelial cells. They suggested that mitochondrial dysfunction activated NF-κB signaling through cGAS-STING recognition, which exacerbated the immune response and contributed to SS pathogenesis [123].

However, despite recent advances in the understanding of SS pathogenesis, many details of the correlation between mitochondrial dysfunction and the immune responses are still unknown. More studies are required to investigate how mitochondrial dysfunction affects immune cell behavior during SS progression and explore whether alleviating mitochondrial dysfunction could contribute to the clinical management of SS.

## 5. Discussion and Conclusions

OIDs affect people’s health and quality of life through damaging oral and craniofacial tissue and increasing the risk of non-oral systemic diseases. Recently, there has ben increasing evidence indicating that mitochondrial dysfunction also plays a critical role in the pathogenesis and progression of OIDs and its associated non-oral systemic diseases such as diabetes and atherosclerosis. On the one hand, previous studies showed that mitochondria play an important role in the innate immune response, as they affect the main pathways involved in the immune response such as toll-like receptors, (NOD)-like receptors and retinoic acid-inducible gene I (RIG-I)-like receptors [124,125,126]. So, when mitochondrial function is disrupted, the release of mtROS, oxidized mtDNA and DAMPs can affect immune responses and aggravate oral inflammatory condition. On the other hand, infection and inflammatory response may also contribute to the disruption of mitochondrial physiological function. For instance, reduced ATP production was observed in both human gingival fibroblast and endothelial cells following *P. gingivalis* and its LPS treatment [77,127]. Verma et al. found that *P. gingivalis*-LPS significantly upregulated the expression of several proinflammatory markers such as iNOS, IL-1β, IL-6 and TNF-α in SH-SY5Y cells and altered the mitochondrial respiration in complex I, II and IV. Mitochondrial genes involved in mitochondria biogenesis, fission and fusion were also downregulated after LPS treatment [15]. A similar phenomenon was also found in immune cells. *P. gingivalis* infection promoted a metabolic shift toward glycolysis and triggered mitochondrial dysfunction in macrophages. These changes were consistent with the alteration in TCA cycle genes and increased glycolytic gene expression [128]. He et al. revealed that under inflammatory conditions, mitochondrial calcium overload in macrophages triggered the persistent opening of mitochondrial permeability transition pores, aggravating calcium overload and inducing mitochondrial dysfunction, forming an adverse cycle which contributed to the activation of periodontitis in vivo [129].

However, current studies are insufficient to describe the exact role of mitochondrial dysfunction in those diseases. For example, whether mitochondrial dysfunction occurs before or after clinical symptoms are shown remains unclear. Several antioxidants such as rutin and resveratrol have shown favorable therapeutic efficacy in alleviating OIDs by regulating mitochondrial membrane potential and improving mitochondrial biogenesis. Nanomaterials designed to target mitochondrial dysfunction also demonstrated huge potential in treating periodontitis. Novel therapeutic approaches, like mitochondria transfer from healthy donor MSCs, have been investigated in treating pulpitis both in vivo and in vitro.

However, the therapeutic strategies mentioned above failed to provide data on the long-term prognosis, and the mechanisms of how improvement in mitochondrial dysfunction affects the prognosis of OIDs were poorly investigated. Therefore, it is vital to characterize the underlying mechanisms by which mitochondrial dysfunction affects OIDs pathogenesis and developing new diagnostic and therapeutic interventions targeting mitochondrial dysfunction in OIDs may be a promising therapeutic strategy in the future.

To sum up, mitochondrial dysfunction has been widely implicated in a wide range of diseases. Disrupted mitochondria integrity and abnormal structure were present in many common oral inflammatory diseases. Several studies have revealed the correlation between mitochondrial dysfunction and the pathogenesis and progression of OIDs, as well as their complications. Further studies should focus on targeting mitochondrial dysfunction in treating those diseases and try to explore the underlying mechanism of how mitochondrial dysfunction affects the occurrence and severity of OIDs, which is a promising field in the future.

## Figures and Tables

**Figure 1 ijms-24-15483-f001:**
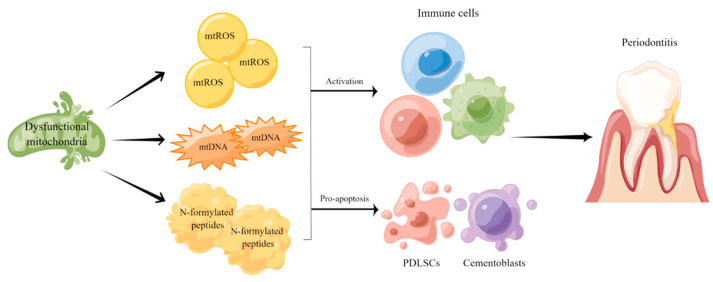
Mitochondrial dysfunction in the pathogenesis and progression of periodontitis. Mitochondrial dysfunction leads to metabolic disruptions and promotes the release of mtROS, mtDNA and N-formylated peptides to regulate immune responses and inflammation in periodontal tissue.

**Table 1 ijms-24-15483-t001:** The role of mitochondrial dysfunction in regulating immune responses in inflammatory diseases.

Origin of Inflammation	Pathogen/Disease	Effect of Mitochondrial Dysfunction on Immune Responses	References
Viral infection	Dengue Virus (DENV)	Mitochondria elongation alleviates DENV-induced retinoic acid inducible gene 1-dependent innate immunity by suppressing interferon-λ1 production, which favors DENV replication.	[34]
	Epstein Barr virus (EBV)	EBV reduces autophagy, decreases intracellular ROS and counteracts mitochondrial biogenesis in differentiating monocytes to prevent the formation of dendritic cells; EBV triggers extensive mitochondrial remodeling and upregulates mitochondrial 1C metabolism in newly infected B cells.	[29,35]
	Herpes simplex virus 1 (HSV-1)	Mitochondria-associated vaccinia virus-related kinase 2 promotes mtDNA release, leading to the cGAS-mediated innate immune response and upregulation of antiviral genes Ifnb1 and Cxcl10.	[36]
	Human immunodeficiency virus (HIV)	mtDNA correlated negatively with inflammatory marker sCD163; platelet mitochondrial function is disturbed in HIV patients, which may contribute to platelet dysfunction and subsequent complications.	[37]
	Severe fever with thrombocytopenia syndrome virus (SFTSV)	SFTSV infection induces mitochondrial damage and mtROS release, triggering excessive inflammatory responses via NLRP3 inflammasome activation.	[38]
Bacterial infection	*Pseudomonas aeruginosa* (*P. aeruginosa*)	In vivo adaptation to high succinate generates *P. aeruginosa* strains which retain the ability to activate mtROS and promote a mitochondrial itaconate response in the airway cells, which suppresses inflammation.	[39]
	*Legionella pneumophila* (*L. pneumophila*)	*L. pneumophila* exerts highly dynamic interactions with host mitochondria by inducing mitochondrial fragmentation and a Warburg-like metabolism in macrophages that favors bacterial replication.	[40]
	*Escherichia coli* and *Neisseria gonorrhoeae*	Outer membrane vesicles from *Escherichia coli* induce mitochondrial apoptosis and NLRP3 inflammasome activation; the activation and release of interleukin-1β in response to *Neisseria gonorrhoeae* OMVs is regulated by mitochondrial apoptosis in vivo.	[41]
Autoimmune factors	Systemic lupus erythematosus (SLE)	Extracellular release of oxidized mitochondrial DNA stimulates type I interferon signaling; mtROS are necessary for neutrophil extracellular traps of low-density granulocytes from SLE patients.	[42]
	Type 1 diabetes	mtDNA activates endothelial NLRP3 inflammasome by Ca^2+^ influx and mtROS generation, which leads to vascular inflammatory damage and endothelial dysfunction.	[43]
	Inflammatory bowel disease	mtDNA is released into the serum and acts as a pro-inflammatory factor and damage-associated molecular pattern (DAMP) for immune cell activation; the release of mtROS and mtDNA into the cytosol are key upstream events in NLRP3 inflammasome activation.	[44,45]
	Atherosclerosis	Lack of mitochondrial uncoupling protein 1 leads to activation of the NLRP3 inflammasome and maturation of interleukin-1β, which exacerbates endothelial dysfunction and vascular inflammation.	[46]
	Rheumatoid arthritis (RA)	Mitochondrial dysfunction induced by tumor necrosis factor-like ligand 1A and tumor necrosis factor receptor 2 increases inflammatory response in RA patients via ROS production.	[47]

## Data Availability

No new data were created.

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
