# Peer review of "Mitochondrial Dysfunction in the Pathogenesis and Treatment of Oral Inflammatory Diseases"

_ijms, 2023, doi:10.3390/ijms242015483_

Round 1

Reviewer 1 Report

Relationships and links between mitochondrial dysfunctions found in this group of incommensurable diseases mentioned in this review, and their more or less known causes, remain very questionable. I.e, the principle question could be what is the cause and what the consequence? What about an association between mitochondrial dysfunction and oral microbiota? In MRONJ, the primary problem lies in an altered quality of the bone of jaws done by impaired blood supply, its ability to heal and regenerate. In SjÅ‘gren´s syndrome believed to be an exemplary systemic disease of autoimmune origin, lots of mechanisms associated with altered functions of immune system has been elucidated. Mitochondrial functions seem to be secondary, secondarily altered. In general, the four diseases with very different etiology, pathogenesis, prognosis, and importance taken as one cluster with certain laboratory similarities regarding mitochondrial dysfunction, detectable very probably in most of inflammatory diseases, does not seem to be suitable idea.

Moderate editing the the languae recommended.

Author Response

Response to reviewer 1

Dear reviewer,

We appreciate your thoughtful and constructive comments. As indicated below, our manuscript has been extensively revised and edited with providing additional information in response to your comments.

Below is the response to address your comments. We have also made corresponding revisions in the text as highlighted with blue text. We hope our responses are reasonable and the revisions we made are acceptable.

Relationships and links between mitochondrial dysfunctions found in this group of incommensurable diseases mentioned in this review, and their more or less known causes, remain very questionable. I.e, the principle question could be what is the cause and what the consequence? What about an association between mitochondrial dysfunction and oral microbiota? In MRONJ, the primary problem lies in an altered quality of the bone of jaws done by impaired blood supply, its ability to heal and regenerate. In SjÅ‘gren´s syndrome believed to be an exemplary systemic disease of autoimmune origin, lots of mechanisms associated with altered functions of immune system has been elucidated. Mitochondrial functions seem to be secondary, secondarily altered. In general, the four diseases with very different etiology, pathogenesis, prognosis, and importance taken as one cluster with certain laboratory similarities regarding mitochondrial dysfunction, detectable very probably in most of inflammatory diseases, does not seem to be suitable idea.

Moderate editing the the languae recommended.

Response:

The aim of this review is to elucidate how mitochondrial dysfunction affects common oral inflammatory diseases including periodontitis, pulpitis, MRONJ and SjÅ‘gren´s syndrome. Although the four diseases are quite different in etiology, pathology, clinical manifestation, treatment and prognosis, we noticed that mitochondrial dysfunction is a common feature in these diseases. Mitochondrial dysfunction is observed in all four OIDs, contributing to their pathogenesis and development.

Regarding the definition of OIDs, we referred to a review published in 2021 by Professor Qianming Chen. In this review, oral mucositis, periodontitis, osteonecrosis of the jaw, Sjögren's syndrome are all included in the range of OIDs, so we also adopt this concept. Another reason we listed these four diseases was that studies have revealed that periodontitis and pulpitis are associated with MRONJ and Sjögren's syndrome. Chuang et al. conducted a population-based retrospective cohort study and found that risk of dental caries, pulpitis, and periodontitis was significantly higher in patients with Sjögren's syndrome. Sroussi et al. suggested that periodontitis may be a trigger of osteoradionecrosis in head and neck cancer patients undergoing radiation therapy. Those patients are also at a higher risk for periodontal disease compared to the general population. Therefore, targeting mitochondrial dysfunction can serve as a promising therapeutic approach to treat the diseases mentioned above, which are still very difficult to treat by clinical practice at present. The studies mentioned above are listed below:

  1. Gong W, Wang F, He Y, Zeng X, Zhang D, Chen Q. Mesenchymal Stem Cell Therapy for Oral Inflammatory Diseases: Research Progress and Future Perspectives. Curr Stem Cell Res Ther. 2021;16(2):165-174. doi:10.2174/1574888X15666200726224132
  2. Chuang CJ, Hsu CW, Lu MC, Koo M. Increased risk of developing dental diseases in patients with primary Sjögren's syndrome-A secondary cohort analysis of population-based claims data. PLoS One. 2020;15(9):e0239442. Published 2020 Sep 18. doi:10.1371/journal.pone.0239442
  3. Sroussi HY, Epstein JB, Bensadoun RJ, et al. Common oral complications of head and neck cancer radiation therapy: mucositis, infections, saliva change, fibrosis, sensory dysfunctions, dental caries, periodontal disease, and osteoradionecrosis. Cancer Med. 2017;6(12):2918-2931. doi:10.1002/cam4.1221

In the review, we elucidated how mitochondrial dysfunction contributes to the pathogenesis and progression of OIDs. But we focused more on introducing novel therapeutic strategies targeting mitochondrial dysfunction in OIDs. We hope that mitochondria-related indicators may assist the diagnosis OIDs and mitochondrial dysfunction can be a common target in developing new treatment for OIDs.

To better elucidate our opinion, we added some content in the discussion and conclusion part (labeled in blue) and we have polished our language as you recommended.

Thank you again for your comments. We really appreciate it.

The added parts are:

Page 7:

Buzoglu et al. collected samples from healthy donors and pulpitis donors to investigate the difference in oxidative stress cycle. They revealed that caries-related inflammatory response altered the oxidative stress cycle in pulp tissue and GSH levels was upregulated due to the increase in ROS levels, which improved the defensive capacity of the dental pulp [99]. Vengerfeldt et al. found higher levels of oxidative stress in pulitis and periodontitis patients were associated with increased dental pain and bone destruction [100]. The results mentioned above revealed how mitochondrial dysfunction and oxidative stress affected pulpitis pathogenesis and its related symptoms.

Page 9:

Xu et al. confirmed that lactate levels were significantly upregulated in the salivary glands of patients with SS and lactate triggered an inflammatory response by damaging mtDNA and causing mtDNA leakage in glandular epithelial cells. They suggested that mitochondrial dysfunction activated NF-κB signaling through cGAS-STING recognition, which exacerbated the immune response and contributed to SS pathogenesis [123].

Page 9:

On the one hand, previous studies showed that mitochondria play an important role in the innate immune response, as they affect main pathways involved in the immune response such as toll-like receptors, (NOD)-like receptors and retinoic acid-inducible gene I (RIG-I)-like receptors [124-126]. So, when mitochondrial function is disrupted, the release of mtROS, oxidized mtDNA and DAMPs can affect immune responses and aggravate oral inflammatory condition. On the other hand, infection and inflammatory response may also contribute to the disruption of mitochondrial physiological function. For instance, reduced ATP production was observed in both human gingival fibroblast and endothelial cells following P. gingivalis and its LPS treatment [77, 127]. Verma et al. found that P. gingivalis-LPS significantly upregulated the expression of several proin-flammatory markers such as iNOS, IL-1β, IL-6 and TNF-α in SH-SY5Y cells and altered the mitochondrial respiration in complex I, II, and IV. Mitochondrial genes involved in mitochondria biogenesis, fission, and fusion were also down regulated after LPS treatment [15]. Similar phenomenon was also found in immune cells. P. gingivalis in-fection promoted a metabolic shift toward glycolysis and triggered mitochondrial dysfunction in macrophages. These changes were consistent to the alteration in TCA cycle genes and increased glycolytic gene expression [128]. He et al. revealed that un-der inflammatory conditions, mitochondrial calcium overload in macrophages trig-gered the persistent opening of mitochondrial permeability transition pores, aggravat-ing calcium overload and inducing mitochondrial dysfunction to form an adverse cycle, which contributed to the activation of periodontitis in vivo [129].

Newly added references:

  1. Gong, W.; Wang, F.; He, Y.; Zeng, X.; Zhang, D.; Chen, Q., Mesenchymal Stem Cell Therapy for Oral Inflammatory Diseases: Research Progress and Future Perspectives. Curr Stem Cell Res Ther 2021, 16, (2), 165-174.

  1. Bitencourt, F. V.; Nascimento, G. G.; Costa, S. A.; Andersen, A.; Sandbæk, A.; Leite, F. R. M., Co-occurrence of Periodontitis and Diabetes-Related Complications. J Dent Res 2023, 102, (10), 1088-1097.

  1. Vujovic, S.; Desnica, J.; Stevanovic, M.; Mijailovic, S.; Vojinovic, R.; Selakovic, D.; Jovicic, N.; Rosic, G.; Milovanovic, D., Oral Health and Oral Health-Related Quality of Life in Patients with Primary Sjögren's Syndrome. Medicina (Kaunas) 2023, 59, (3).

  1. Sødal, A. T. T.; Skudutyte-Rysstad, R.; Diep, M. T.; Koldsland, O. C.; Hove, L. H., Periodontitis in a 65-year-old population: risk indicators and impact on oral health-related quality of life. BMC Oral Health 2022, 22, (1), 640.

  1. Taha, N. A.; Abuzaid, A. M.; Khader, Y. S., A Randomized Controlled Clinical Trial of Pulpotomy versus Root Canal Therapy in Mature Teeth with Irreversible Pulpitis: Outcome, Quality of Life, and Patients' Satisfaction. J Endod 2023, 49, (6), 624-631.e2.

  1. Jiang, W.; Wang, Y.; Cao, Z.; Chen, Y.; Si, C.; Sun, X.; Huang, S., The role of mitochondrial dysfunction in periodontitis: From mechanisms to therapeutic strategy. J Periodontal Res 2023, 58, (5), 853-863.

  1. Verma, A.; Azhar, G.; Zhang, X.; Patyal, P.; Kc, G.; Sharma, S.; Che, Y.; Wei, J. Y., P. gingivalis-LPS Induces Mitochondrial Dysfunction Mediated by Neuroinflammation through Oxidative Stress. Int J Mol Sci 2023, 24, (2).

  1. Zhou, L.; Zhang, Y. F.; Yang, F. H.; Mao, H. Q.; Chen, Z.; Zhang, L., Mitochondrial DNA leakage induces odontoblast inflammation via the cGAS-STING pathway. Cell Commun Signal 2021, 19, (1), 58.

  1. Dogan Buzoglu, H.; Ozcan, M.; Bozdemir, O.; Aydin Akkurt, K. S.; Zeybek, N. D.; Bayazit, Y., Evaluation of oxidative stress cycle in healthy and inflamed dental pulp tissue: a laboratory investigation. Clin Oral Investig 2023, 27, (10), 5913-5923.

  1. Vengerfeldt, V.; Mändar, R.; Saag, M.; Piir, A.; Kullisaar, T., Oxidative stress in patients with endodontic pathologies. J Pain Res 2017, 10, 2031-2040.

  1. Xu, J.; Chen, C.; Yin, J.; Fu, J.; Yang, X.; Wang, B.; Yu, C.; Zheng, L.; Zhang, Z., Lactate-induced mtDNA Accumulation Activates cGAS-STING Signaling and the Inflammatory Response in Sjögren's Syndrome. Int J Med Sci 2023, 20, (10), 1256-1271.

  1. Costa, T. J.; Potje, S. R.; Fraga-Silva, T. F. C.; da Silva-Neto, J. A.; Barros, P. R.; Rodrigues, D.; Machado, M. R.; Martins, R. B.; Santos-Eichler, R. A.; Benatti, M. N.; de Sá, K. S. G.; Almado, C. E. L.; Castro Í, A.; Pontelli, M. C.; Serra, L.; Carneiro, F. S.; Becari, C.; Louzada-Junior, P.; Oliveira, R. D. R.; Zamboni, D. S.; Arruda, E.; Auxiliadora-Martins, M.; Giachini, F. R. C.; Bonato, V. L. D.; Zachara, N. E.; Bomfim, G. F.; Tostes, R. C., Mitochondrial DNA and TLR9 activation contribute to SARS-CoV-2-induced endothelial cell damage. Vascul Pharmacol 2022, 142, 106946.

  1. Andrieux, P.; Chevillard, C.; Cunha-Neto, E.; Nunes, J. P. S., Mitochondria as a Cellular Hub in Infection and Inflammation. Int J Mol Sci 2021, 22, (21).

  1. Mills, E. L.; Kelly, B.; O'Neill, L. A. J., Mitochondria are the powerhouses of immunity. Nat Immunol 2017, 18, (5), 488-498.

  1. Napa, K.; Baeder, A. C.; Witt, J. E.; Rayburn, S. T.; Miller, M. G.; Dallon, B. W.; Gibbs, J. L.; Wilcox, S. H.; Winden, D. R.; Smith, J. H.; Reynolds, P. R.; Bikman, B. T., LPS from P. gingivalis Negatively Alters Gingival Cell Mitochondrial Bioenergetics. Int J Dent 2017, 2017, 2697210.

  1. Fleetwood, A. J.; Lee, M. K. S.; Singleton, W.; Achuthan, A.; Lee, M. C.; O'Brien-Simpson, N. M.; Cook, A. D.; Murphy, A. J.; Dashper, S. G.; Reynolds, E. C.; Hamilton, J. A., Metabolic Remodeling, Inflammasome Activation, and Pyroptosis in Macrophages Stimulated by Porphyromonas gingivalis and Its Outer Membrane Vesicles. Front Cell Infect Microbiol 2017, 7, 351.

  1. He, P.; Liu, F.; Li, M.; Ren, M.; Wang, X.; Deng, Y.; Wu, X.; Li, Y.; Yang, S.; Song, J., Mitochondrial Calcium Ion Nanogluttons Alleviate Periodontitis via Controlling mPTPs. Adv Healthc Mater 2023, 12, (15), e2203106.

Reviewer 2 Report

Dear authors, this paper appears well-written and very simple to read, even if the reader is not an expert in the biological field.

There are some suggestions for you in order to gain better scientific soundness

-       The abstract is well organized but lacks result and conclusion sections

-       Lines 40-51, please add references

-       Please focus more on the role of mitochondria dysfunction in the etiopathogenesis of the OIDs, as what is reported in the present literature and what we must know better on such topic.

-       Please add a material and methods section where you can explain how you conducted this review (which keywords you searched for or which database you used)

-       Line 97, there is a formatting error: please correct in membrane

-       Please split the periodontitis section into two different sections: the first as “the role of mitochondria dysfunction in the etiopathogenesis of the chronic periodontitis” and “the mitochondrial dysfunction targeted therapies”

-       Please move Fig. 1 under the role of mitochondria dysfunction in the etiopathogenesis of the chronic periodontitis section, citing it in the main text.

-  Please merge the discussion and conclusion sections into one

However, this review is well-written and readable, even if it is almost incomplete, because it considers only a few diseases that belong to the wide range of OIDs. But I have a question that is mandatory to explain better in the main text: is well defined and reported the influence of mitochondrial dysfunction in the inflammatory disorder, but contrariwise, the inflammatory and infectious disease could have an impact on the mitochondrial physiological function? Here, I report to you a review (to consider as an example) where the authors described the influence of the infection on mitochondrial functions in a very analytical way. Have you assumed the role of inflammation or infections (in the cases of periodontitis or pulpitis) on mitochondrial functions? Please address this issue to improve the scientific relevance of your work

Andrieux P, Chevillard C, Cunha-Neto E, Nunes JPS. Mitochondria as a Cellular Hub in Infection and Inflammation. Int J Mol Sci. 2021 Oct 20;22(21):11338. doi: 10.3390/ijms222111338. PMID: 34768767; PMCID: PMC8583510.

Nice paper, well done. 

Author Response

Response to reviewer 2

Dear reviewer,

We appreciate your thoughtful and constructive comments. As indicated below, our manuscript has been extensively revised and edited with providing additional information in response to your comments.

Below is the response to address your comments. We have also made corresponding revisions in the text as highlighted with blue text. We hope our responses are reasonable and the revisions we made are acceptable.

-       The abstract is well organized but lacks result and conclusion sections

Response: We have re-organized the abstract section and added result and conclusion sections.

-       Lines 40-51, please add references

Response: We have added references according to the content of line 40-51.

-       Please focus more on the role of mitochondria dysfunction in the etiopathogenesis of the OIDs, as what is reported in the present literature and what we must know better on such topic.

Response: We have added some content regarding the role of mitochondrial dysfunction in the etiopathogenesis of the OIDs in the discussion and conclusions section as a summary. Latest research on such topic were added and discussed. The added parts are:

Page 7:

Buzoglu et al. collected samples from healthy donors and pulpitis donors to investigate the difference in oxidative stress cycle. They revealed that caries-related inflammatory response altered the oxidative stress cycle in pulp tissue and GSH levels was upregulated due to the increase in ROS levels, which improved the defensive capacity of the dental pulp [99]. Vengerfeldt et al. found higher levels of oxidative stress in pulitis and periodontitis patients were associated with increased dental pain and bone destruction [100]. The results mentioned above revealed how mitochondrial dysfunction and oxidative stress affected pulpitis pathogenesis and its related symptoms.

Page 9:

Xu et al. confirmed that lactate levels were significantly upregulated in the salivary glands of patients with SS and lactate triggered an inflammatory response by damaging mtDNA and causing mtDNA leakage in glandular epithelial cells. They suggested that mitochondrial dysfunction activated NF-κB signaling through cGAS-STING recognition, which exacerbated the immune response and contributed to SS pathogenesis [123].

-       Please add a material and methods section where you can explain how you conducted this review (which keywords you searched for or which database you used)

Response: We have added a material and methods section in the abstract section to explain the searching strategy of this review.

-       Line 97, there is a formatting error: please correct in membrane

Response: The formatting error has been addressed.

-       Please split the periodontitis section into two different sections: the first as “the role of mitochondria dysfunction in the etiopathogenesis of the chronic periodontitis” and “the mitochondrial dysfunction targeted therapies”

Response: We have split the periodontitis section into two sections accordingly.

-       Please move Fig. 1 under the role of mitochondria dysfunction in the etiopathogenesis of the chronic periodontitis section, citing it in the main text.

Response: We have moved Figure. 1 under the role of mitochondria dysfunction in the etiopathogenesis of the chronic periodontitis section and cited in the main text.

-  Please merge the discussion and conclusion sections into one

Response: We have merged the discussion and conclusion sections into one section.

However, this review is well-written and readable, even if it is almost incomplete, because it considers only a few diseases that belong to the wide range of OIDs. But I have a question that is mandatory to explain better in the main text: is well defined and reported the influence of mitochondrial dysfunction in the inflammatory disorder, but contrariwise, the inflammatory and infectious disease could have an impact on the mitochondrial physiological function? Here, I report to you a review (to consider as an example) where the authors described the influence of the infection on mitochondrial functions in a very analytical way. Have you assumed the role of inflammation or infections (in the cases of periodontitis or pulpitis) on mitochondrial functions? Please address this issue to improve the scientific relevance of your work

Response: Thank you for your constructive advice. We have read the review you recommended very carefully. We added the following part in the discussion section to further elucidate how inflammation and infection can affect the mitochondrial function in OIDs.

The added part:

“On the one hand, previous studies showed that mitochondria play an important role in the innate immune response, as they affect main pathways involved in the immune response such as toll-like receptors, (NOD)-like receptors and retinoic acid-inducible gene I (RIG-I)-like receptors [121-123]. So, when mitochondrial function is disrupted, the release of mtROS, oxidized mtDNA and DAMPs can affect immune responses and aggravate oral inflammatory condition. On the other hand, infection and inflammatory response may also contribute to the disruption of mitochondrial physiological function. For instance, reduced ATP production was observed in both human gingival fibroblast and endothelial cells following P. gingivalis and its LPS treatment [77, 124]. Verma et al. found that P. gingivalis-LPS significantly upregulated the expression of several proin-flammatory markers such as iNOS, IL-1β, IL-6 and TNF-α in SH-SY5Y cells and altered the mitochondrial respiration in complex I, II, and IV. Mitochondrial genes involved in mitochondria biogenesis, fission, and fusion were also down regulated after LPS treatment [15]. Similar phenomenon was also found in immune cells. P. gingivalis in-fection promoted a metabolic shift toward glycolysis and triggered mitochondrial dysfunction in macrophages. These changes were consistent to the alteration in TCA cycle genes and increased glycolytic gene expression [125]. He et al. revealed that un-der inflammatory conditions, mitochondrial calcium overload in macrophages trig-gered the persistent opening of mitochondrial permeability transition pores, aggravat-ing calcium overload and inducing mitochondrial dysfunction to form an adverse cycle, which contributed to the activation of periodontitis in vivo [126].”

Newly added references:

  1. Gong, W.; Wang, F.; He, Y.; Zeng, X.; Zhang, D.; Chen, Q., Mesenchymal Stem Cell Therapy for Oral Inflammatory Diseases: Research Progress and Future Perspectives. Curr Stem Cell Res Ther 2021, 16, (2), 165-174.

  1. Bitencourt, F. V.; Nascimento, G. G.; Costa, S. A.; Andersen, A.; Sandbæk, A.; Leite, F. R. M., Co-occurrence of Periodontitis and Diabetes-Related Complications. J Dent Res 2023, 102, (10), 1088-1097.

  1. Vujovic, S.; Desnica, J.; Stevanovic, M.; Mijailovic, S.; Vojinovic, R.; Selakovic, D.; Jovicic, N.; Rosic, G.; Milovanovic, D., Oral Health and Oral Health-Related Quality of Life in Patients with Primary Sjögren's Syndrome. Medicina (Kaunas) 2023, 59, (3).

  1. Sødal, A. T. T.; Skudutyte-Rysstad, R.; Diep, M. T.; Koldsland, O. C.; Hove, L. H., Periodontitis in a 65-year-old population: risk indicators and impact on oral health-related quality of life. BMC Oral Health 2022, 22, (1), 640.

  1. Taha, N. A.; Abuzaid, A. M.; Khader, Y. S., A Randomized Controlled Clinical Trial of Pulpotomy versus Root Canal Therapy in Mature Teeth with Irreversible Pulpitis: Outcome, Quality of Life, and Patients' Satisfaction. J Endod 2023, 49, (6), 624-631.e2.

  1. Jiang, W.; Wang, Y.; Cao, Z.; Chen, Y.; Si, C.; Sun, X.; Huang, S., The role of mitochondrial dysfunction in periodontitis: From mechanisms to therapeutic strategy. J Periodontal Res 2023, 58, (5), 853-863.

  1. Verma, A.; Azhar, G.; Zhang, X.; Patyal, P.; Kc, G.; Sharma, S.; Che, Y.; Wei, J. Y., P. gingivalis-LPS Induces Mitochondrial Dysfunction Mediated by Neuroinflammation through Oxidative Stress. Int J Mol Sci 2023, 24, (2).

  1. Zhou, L.; Zhang, Y. F.; Yang, F. H.; Mao, H. Q.; Chen, Z.; Zhang, L., Mitochondrial DNA leakage induces odontoblast inflammation via the cGAS-STING pathway. Cell Commun Signal 2021, 19, (1), 58.

  1. Dogan Buzoglu, H.; Ozcan, M.; Bozdemir, O.; Aydin Akkurt, K. S.; Zeybek, N. D.; Bayazit, Y., Evaluation of oxidative stress cycle in healthy and inflamed dental pulp tissue: a laboratory investigation. Clin Oral Investig 2023, 27, (10), 5913-5923.

  1. Vengerfeldt, V.; Mändar, R.; Saag, M.; Piir, A.; Kullisaar, T., Oxidative stress in patients with endodontic pathologies. J Pain Res 2017, 10, 2031-2040.

  1. Xu, J.; Chen, C.; Yin, J.; Fu, J.; Yang, X.; Wang, B.; Yu, C.; Zheng, L.; Zhang, Z., Lactate-induced mtDNA Accumulation Activates cGAS-STING Signaling and the Inflammatory Response in Sjögren's Syndrome. Int J Med Sci 2023, 20, (10), 1256-1271.

  1. Costa, T. J.; Potje, S. R.; Fraga-Silva, T. F. C.; da Silva-Neto, J. A.; Barros, P. R.; Rodrigues, D.; Machado, M. R.; Martins, R. B.; Santos-Eichler, R. A.; Benatti, M. N.; de Sá, K. S. G.; Almado, C. E. L.; Castro Í, A.; Pontelli, M. C.; Serra, L.; Carneiro, F. S.; Becari, C.; Louzada-Junior, P.; Oliveira, R. D. R.; Zamboni, D. S.; Arruda, E.; Auxiliadora-Martins, M.; Giachini, F. R. C.; Bonato, V. L. D.; Zachara, N. E.; Bomfim, G. F.; Tostes, R. C., Mitochondrial DNA and TLR9 activation contribute to SARS-CoV-2-induced endothelial cell damage. Vascul Pharmacol 2022, 142, 106946.

  1. Andrieux, P.; Chevillard, C.; Cunha-Neto, E.; Nunes, J. P. S., Mitochondria as a Cellular Hub in Infection and Inflammation. Int J Mol Sci 2021, 22, (21).

  1. Mills, E. L.; Kelly, B.; O'Neill, L. A. J., Mitochondria are the powerhouses of immunity. Nat Immunol 2017, 18, (5), 488-498.

  1. Napa, K.; Baeder, A. C.; Witt, J. E.; Rayburn, S. T.; Miller, M. G.; Dallon, B. W.; Gibbs, J. L.; Wilcox, S. H.; Winden, D. R.; Smith, J. H.; Reynolds, P. R.; Bikman, B. T., LPS from P. gingivalis Negatively Alters Gingival Cell Mitochondrial Bioenergetics. Int J Dent 2017, 2017, 2697210.

  1. Fleetwood, A. J.; Lee, M. K. S.; Singleton, W.; Achuthan, A.; Lee, M. C.; O'Brien-Simpson, N. M.; Cook, A. D.; Murphy, A. J.; Dashper, S. G.; Reynolds, E. C.; Hamilton, J. A., Metabolic Remodeling, Inflammasome Activation, and Pyroptosis in Macrophages Stimulated by Porphyromonas gingivalis and Its Outer Membrane Vesicles. Front Cell Infect Microbiol 2017, 7, 351.

  1. He, P.; Liu, F.; Li, M.; Ren, M.; Wang, X.; Deng, Y.; Wu, X.; Li, Y.; Yang, S.; Song, J., Mitochondrial Calcium Ion Nanogluttons Alleviate Periodontitis via Controlling mPTPs. Adv Healthc Mater 2023, 12, (15), e2203106.

Round 2

Reviewer 1 Report

 acceptable now, thank you.

Reviewer 2 Report

Dear authors, thank you for such consideration of my comments.

Each point was addressed, but there are some minor changes.

-       The abstract is now well organized, but there should be no spaces between different sections. The abstract is a single text without space between two lines. Thanks

-       Line 255 please indicate the meaning of “GSH”

-       Material and methods must be reported in the main text. The abstract is only a summary of the main text. Please add material and methods section in the main text after the introduction section, following MDPI guidelines.

The other comments were adequately addressed. 

Thank you and good luck

Author Response

Response to reviewer 2

Dear reviewer,

We appreciate your thoughtful and constructive comments. As indicated below, our manuscript has been revised and edited according to your comments.

Below is the response to address your comments. We have also made corresponding revisions in the text as highlighted with blue text. We hope our responses are reasonable and the revisions we made are acceptable.

-       The abstract is now well organized, but there should be no spaces between different sections. The abstract is a single text without space between two lines. Thanks

Response: We have deleted the spaces between different sections in the abstract.

-       Line 255 please indicate the meaning of “GSH”

Response: We have indicated the meaning of GSH in the main text.

-       Material and methods must be reported in the main text. The abstract is only a summary of the main text. Please add material and methods section in the main text after the introduction section, following MDPI guidelines.

Response: We have moved the material and methods section below the introduction section and adjusted the numbers accordingly.

The other comments were adequately addressed. 

Response: Thank you for your time and comments. We really appreciate your help.